# Cognitive Function and the Consumption of Probiotic Foods: A National Health and Nutrition Examination Survey Study

**DOI:** 10.3390/nu16213631

**Published:** 2024-10-25

**Authors:** Lora J. Kasselman, Morgan R. Peltier, Joshua De Leon, Allison B. Reiss

**Affiliations:** 1Department of Medical Sciences, Hackensack Meridian School of Medicine, Nutley, NJ 07110, USA; lora.kasselman@hmhn.org; 2Hackensack Meridian Health Research Institute, Hackensack, NJ 07601, USA; 3Department of Psychiatry, Hackensack Meridian School of Medicine, Nutley, NJ 07110, USA; morgan.peltier@hmnh.org; 4Department of Psychiatry, Jersey Shore University Medical Center, Neptune City, NJ 07753, USA; 5Department of Medicine, NYU Grossman Long Island School of Medicine, Mineola, NY 11501, USA; joshua.deleon@nyulangone.org

**Keywords:** cognitive function, microbiome, dairy, diet

## Abstract

**Background/Objectives:** Impaired cognition is a key trait of the diseases of aging and is an important quality of life factor for older adults and their families. Over the past decade, there has been an increasing appreciation for the role of the microbiome in cognition, as well as emerging evidence that probiotics, such as those in yogurt and other dairy products, can have a positive impact on cognitive function. However, it is unclear to what extent the consumption of yogurt is associated with improved cognitive function in older adults. **Methods:** Therefore, we compared the scores for the Wechsler Adult Intelligence Scale, Digit–Symbol Substitution Test between respondents who self-reported daily yogurt/dairy consumption with those who claimed they did not in an NHANES. **Results:** We found that cognitive scores were significantly higher (40.03 ± 0.64 vs. 36.28 ± 1.26, *p* = 0.017) in respondents reporting daily yogurt/dairy consumption, though only a trend remained after adjusting for sociodemographic covariates (*p* = 0.074). **Conclusions:** Further studies are required to confirm that this is a cause–effect relationship and whether changing diets is a low-cost means of protecting aging populations from cognitive decline and improving their quality of life.

## 1. Introduction

Cognition, the process by which humans acquire knowledge and understanding through sense experience, is fundamentally important for quality of life [1]. Many factors contribute to cognitive function, including genetic and environmental factors such as overall health, nutrient availability, mental health, development, and education [2]. Cognitive impairment can affect a person’s ability to remember, learn, concentrate, or make decisions and can range from mild to severe, possibly affecting their ability to function independently [3]. Cognitive impairment is typically associated with dementia but can also present across multiple diseases, including depression, autoimmunity, and post-traumatic stress disorder [4]. Recent census data indicate a prevalence of 22.7 cases of mild cognitive impairment (MCI) per 100 in the United States (US) for all adults 65 years and older [5], and costs associated with dementia-related cognitive impairment, including unpaid family provided care, is estimated at over USD 100 billion per year [6]. Worldwide, the median prevalence of MCI in community-dwelling adults over the age of 50 has been estimated to be between 17% and 19%, indicating that this is also a global issue [7,8].

Treatment for cognitive dysfunction is difficult and usually dependent upon treating the underlying cause such as diabetes or hypertension, where there is often limited success [9,10]. The standard of care for cognitively impaired patients is highly variable and person-dependent [11]. Patients should receive goal-directed care that reduces other risks, such as falls, and that is appropriate with respect to their underlying comorbidities, such as diabetes [11]. Treatment for underlying disease processes may be associated with numerous side effects, stigma, and high costs that result in non-compliance, presenting a significant challenge for clinical management [9,10,12]. Although non-pharmacological compounds, such as probiotics, are not the standard of care [13] for treating cognitive impairment, recent studies have shown that they may modulate the gut microbiome [14], improve learning, reduce underlying anxiety and depressive symptoms, and reduce negative thoughts associated with sadness [15,16,17]. Probiotic consumption also modulates neurotransmitter and growth factor levels in the brain [18], suggesting a mechanism of action.

While small-scale animal and human studies suggest that the gut microbiome is important in regulating mental health, few have looked at population-level associations between probiotics consumption and cognition [19]. The impact of probiotics on cognition is an important area to investigate as it may indicate a low-cost, easily complied with, stand alone, or supplemental treatment for cognitive dysfunction to improve quality of life [20].

The objective of this study is to determine whether the consumption of probiotics is related to increased measures of cognitive function in a representative national sample. Our hypothesis is that people who report consuming probiotics will perform better on a cognitive function test than those who do not consume probiotics in a representative national sample. Study of this research question is critical because if this hypothesis is true, it could expand opportunities to use probiotic supplementation in humans to treat and/or prevent cognitive dysfunction across different diseases such as depression, autoimmunity, dementia, and PTSD.

## 2. Materials and Methods

### 2.1. Study Population

The National Health and Nutrition Examination Survey (NHANES) is a population-based survey designed to assess the health and nutritional status of adults and children in the US. The survey examines a nationally representative sample of about 5000 persons each year and includes information on demographic, socioeconomic, dietary, and health-related variables. NHANES 1999–2000 was conducted between 1999 and 2000 and contains over 10,000 respondents, including children and people from under-represented groups [21]. For the current study, data on probiotic consumption behavior, cognitive function, sociodemographic variables, and other characteristics were extracted.

### 2.2. Probiotic Consumption Behavior

Interview data were extracted, including the type of foods consumed by each participant. Consumption of the probiotic food yogurt was included in the NHANES Dietary Behavior and Nutrition Questionnaire (DBQ) and was combined with other types of dairy consumption, including milk and cottage cheese but not frozen yogurt. The interviewer asked one time per participant how many helpings per day the participant ate of the above-mentioned foods. If the participant reported more than zero times per day, they were coded as “yes” for probiotic consumption and “no” if they did not.

### 2.3. Cognitive Function

Cognitive function data from NHANES 1999–2000 was limited to adults 60 years of age or older, so data were subsetted to include only those ages for analysis. The Wechsler Adult Intelligence Scale, Third Edition Digit–Symbol Substitution Test (DSST) was used to measure cognitive function. The DSST requires participants to match symbols to numbers based on a provided key, and the correct matches are summed [22]. A higher DSST score indicates higher cognitive function.

### 2.4. Sociodemographic Characteristics

Data on age, sex, race and ethnicity, highest level of education, and poverty-to-income ratio were extracted. Self-reported race and ethnicity was classified into one of five racial/ethnic groups: Non-Hispanic White, Non-Hispanic Black, Mexican–American, Other Hispanic, and Multi-Racial or Other Race. Participant’s education levels were classified into five groups: less than 9th grade, 9–11th grade (including 12th grade with no diploma), high school graduate/General Education Diploma (GED) equivalent, some college or associate’s degree, and college graduate or above.

### 2.5. Body Mass Index (BMI)

Weight and height were measured at the time of physical examination by trained technicians using standardized equipment, and BMI was calculated as weight in kg/(height in meters)^2^.

### 2.6. Statistical Analysis

Survey analysis procedures were used to account for sample weights, stratification, and clustering that was a part of the complex sampling design used to ensure nationally representative estimates. Two-year weights were used based on the variable with the smallest sample size, DSST. NHANES participants aged 60 and older with complete information on probiotics consumption, cognitive function, and other characteristics were included in the analyses. Summarized weighted means and standard errors, along with weighted number and percentages, were calculated for continuous and categorical variables, respectively. Linear regressions were carried out to quantify associations between probiotics consumption and cognitive function test scores. Multivariable linear regression models were adjusted for age, sex, race, BMI, education level, and poverty-to-income ratio. Residuals from the fit models were examined for normality, homoskedasticity, and independence. All statistical analyses were performed using R (https://www.R-project.org/ version 4.3.3, accessed on 4 August 2024), and results where *p* < 0.05 were considered statistically significant.

### 2.7. Estimation of Cost Savings for Potential Interventions to Delay Onset of MCI

Utilizing the latest US Census data for incidence of MCI, we calculated the economic burden in US dollars based on published cost estimates of MCI and dementia [5,23]. We then calculated savings for delaying the onset of MCI based on the US Census numbers for MCI, the American Medical Association numbers for the US population of older adults, and published savings estimates [24,25].

### 2.8. Ethics Approval of Research

The NHANES protocols were approved by the National Center for Health Statistics (NCHS) ethics review board. This study did not require further institutional review as this was an analysis of secondary data without personal identifiers.

## 3. Results

The NHANES 1999–2000 sample had 9965 participants. Of these participants, only 1834 were 60 years or older. Out of this sub-group of older adult NHANES 1999–20,000 participants, there were 1412 that completed both the cognitive function test and answered the question about yogurt/dairy consumption. Of those, 1079 had all of their sociodemographic information available for analysis (Figure 1). Based on a power calculation using our final model, 1079 participants are more than sufficient to achieve power >0.80. The average age of the participants was 70.1 ± 0.4 years (Figure 2). The WAIS III Digit–Symbol results were normally distributed, with an average score of 45.4 ± 0.9 correct, with lower scores indicating lower cognitive function. The weighted demographic characteristics of those who completed the WAIS III Digit–Symbol test are presented in Table 1, stratified by those who reported daily yogurt/dairy consumption and those who reported no daily yogurt/dairy consumption. Several models were used to regress WAIS III scores on yogurt/dairy consumption with the inclusion or exclusion of different covariates or individual covariates alone (Table 2). Daily consumption of yogurt/dairy was positively associated with WAIS III scores in an unadjusted model (β = 4.74, 95% CI: 0.98 to 8.50; *p* = 0.02, Table 2). The full WAIS III model scores regressed on dichotomized yogurt/dairy consumption, including age, gender, race/ethnicity, education, poverty-to-income ratio, and BMI, identified only age, education, and poverty-to-income ratio as significant covariates, so the other covariates were dropped in the final model. After adjusting for age, education, and poverty-to-income ratio, there was a trend where daily consumption of yogurt/dairy was positively associated with WAIS III scores (β = 2.43, 95% CI: −0.31 to 5.16, *p* = 0.074, Table 2). This trend shows that those who reported daily yogurt/dairy consumption trended towards higher WAIS III scores (mean = 46.19, standard error = 0.87) compared to those who did not report daily yogurt/dairy consumption (mean = 41.45; standard error = 1.79).

The 2020 US Census data showed 22.7 cases of MCI per 100 adults of 65 years and older [5]. The US Census Bureau estimates that the number of people over the age of 65 in the US is 54 million [23]. Based on these numbers, there are about 12,250,000 persons with MCI in the US. The cost of care for persons with MCI is estimated at USD 33,700 per year, which is less than the estimated cost of care per person per year for dementia, at USD 51,000 [25]. A range of about 8–17% of people with MCI progress to dementia each year [24,26,27]. As of 2022, about 4% of persons over the age of 65 in the US have received a dementia diagnosis [28]. For this study, we chose a 10% yearly rate of progression for ease of calculations. If even 1% (about 12,000) of people with MCI who would have progressed to dementia could be delayed in their progression to dementia or prevented from progressing to dementia entirely, the savings created by that 1% not progressing is approximately USD 212 million per year (Table 3).

## 4. Discussion

This study identified a trend towards an association between yogurt/dairy consumption and cognitive function scores in older adults, where those who reported consuming daily yogurt or dairy products had a higher cognitive function score. The cognitive function scores were adjusted by age, education, and income level as strong confounders in both the exposure (consumption behavior) and outcome (cognitive function). The difference in the weighted average cognitive function score by yogurt/dairy consumption was 4.74 points, as measured by the Digit–Symbol Substitution Test (DSST).

The DSST is a reliable and valid test that measures cognitive function and is sensitive to cognitive deficits across many conditions. The DSST measures motor speed, attention, visual perception, writing ability, associative learning, and executive function, including working memory [22,29]. Studies using the DSST have shown that a four-point difference in DSST score is clinically relevant and can translate to meaningful improvements in activities of daily living [30,31]. Maintenance of cognition is important, especially in older adults, since decline in cognitive function is a major predictor of dementia [32]. Therefore, the current results suggest that a simple dietary modification, i.e., the consumption of daily yogurt/dairy, could potentially help older adults maintain cognitive function and, consequently, their independence.

Probiotics commonly found in yogurt include Lactobacillus strains, specifically *Lactobacillus delbrueckii* subsp. *bulgaricus* and *Lactobacillus acidophilus Streptococcus thermophilus* [33,34,35]. Cheeses contain primarily Lactobacillus strains and have been proposed as a vehicle for convenient consumption of other health-promoting bacterial species [36,37,38]. Cheeses that have been aged are best, but if they have been heated, the bacteria likely will not survive. In cottage cheese specifically, *Lactobacillus plantarum* has been detected [39].

Yogurt contains probiotic bacteria, such as Bifidobacteria, which promote a healthy gut microbiome [40]. Recent research has begun to identify the important role that the gut microbiome plays in cognitive function and how dysregulation in gut flora is associated with cognitive dysfunction [41,42,43,44]. In a randomized controlled trial in community-dwelling older adults, the consumption of Bifidobacterium supplements for 12 weeks led to changes in the gut microbiome and improvement in mental flexibility scores compared to placebo controls [45]. A meta-analysis of probiotics supplementation in older adults showed that probiotics can significantly improve cognition in both cognitively impaired and Alzheimer’s dementia patients [46], further supporting the therapeutic potential of probiotic foods such as yogurt. Another recent study linked fermented dairy to better executive function and verbal fluency in older persons [47]. However, a study from Sweden of 1334 adults with median age 67, followed for a median of 5 years, showed no significant effect of self-reported level of dairy product consumption on cognition [48].

A number of studies have documented the effect of dairy on the microbial flora of the gut. Swarte et al. conducted a randomized cross-over study in healthy, overweight males and postmenopausal females between the ages of 45 and 65 years on a high-dairy versus a low-dairy diet over 6 weeks with a 4-week washout in between [49]. They compared fecal microbiome composition in 46 subjects to see the differences based on the level of dairy consumption and found numerous differences. The high-dairy diet brought about a decrease in some butyrate-producing bacteria, mainly *Faecalibacterium prausnitzii*, as well as *Eubacterium rectale* and *Roseburia* spp., and the genera *Faecalibacterium* and *Bilophila* and a possible change in the pathway of butyrate production. High dairy resulted in greater abundance of *Streptococcus thermophilus*, *Erysipelatoclostridium ramosum*, and *Leuconostoc mesenteroides. Streptococcus*, *Leuconostoc*, and *Lactococcus* were significantly increased with high-dairy compared to the low-dairy intake. Overall diversity of the microbiome was not altered. A systemic review from Aslam et al. [50] found that milk, yogurt, and kefir increased *Lactobacillus* and *Bifidobacterium*.

It has been postulated that the benefits of *Lactobacilli* and *Bifidobacteria* to the gut–brain axis may be mediated by their anti-inflammatory properties and by their production of gamma-aminobutyric acid, a bioactive compound that functions as an inhibitory neurotransmitter in the central nervous system [51,52,53,54]. In a placebo-controlled study, Lew et al. found that 12 weeks of *Lactobacillus plantarum* ingestion by adults aged 18–60 with moderate stress levels reduced stress and improved social and emotional cognition in women and basic attention in women, while improving verbal learning and memory in men [55]. It also protected memory in persons over age 65 years with memory deterioration [56]. This effect has been attributed to reduced inflammation and oxidative stress with changes in gut microbiota based on evidence from mouse models [57,58]. Zhang et al. found positive effects on cognitive function in an Alzheimer’s disease mouse model given *Streptococcus thermophilus* via gavage for 12 weeks. The cognitive improvement was accompanied by a reduced astrocyte number in the brain [59]. However, the effects of individual microbes on cognition are not yet fully known, and clinically relevant future studies will likely need to consider them in aggregate rather than individually [42,60,61].

The importance of the intestinal microbiome in overall health is increasingly being recognized and many studies are concentrating on this relationship, but in humans, we are limited by the difficulty in finding out the exact food consumed, especially on a long-term basis, and by the many other influences that can affect diet and health [62,63]. Identifying ways to balance the gut bacterial composition would ideally include an analysis of the change in fecal bacterial composition with dietary manipulation, but this is not yet practical on a large scale, and further, the stool microbiome does not dependably mirror the microbiome of the intestinal mucosa [64,65,66]. A study by Shuai et al. was able to look at fecal samples. The authors examined the influence of dairy food ingestion and gut microbiota in the context of cardiometabolic disease risk using data from the Guangzhou Nutrition and Health Study [67,68]. The longitudinal data of a cohort of adults aged 45 to 70 living in Gaungzhou, China, included DNA extraction from stool and categorization of dairy consumption based on servings per week using a validated food frequency questionnaire. They found that the genera in greater abundance in the feces of those with higher dairy consumption were *Bifidobacterium*, *Streptococcus*, *Clostridium*, and *Gemellaceae*, while a genus from the Enterobacteriaceae family was elevated in those who ate less dairy. *Roseburia*, *Lachnobacterium*, *Megasphaera*, and several others were elevated in consumers of larger amounts of yogurt. The ingestion of all dairy and the ingestion of yogurt specifically were both associated with a greater diversity of gut microbiota. One mechanism proposed to explain brain health promotion based on changes in gut flora is the production of butyrate by genera such as *Clostridium*, *Lachnobacterium*, and *Roseburia*. Butyrate has been shown in animal and cell culture models to be neuroprotective through its support of mitochondrial activity and its alleviation of oxidative stress and neuroinflammation [69,70,71]. Butyrate may also impede amyloid formation, known to be neurotoxic and a hallmark of Alzheimer’s disease [72,73,74].

Evidence from the current study adds to the growing literature regarding the pro-cognitive effect of probiotic foods such as yogurt [75,76,77]. However, there are several limitations to our study. Although we adjusted for several known confounders of dietary consumption and cognition, the impact of other unknown, unmeasured confounders on the observed association is unclear. The data used in this study are from a cross-sectional NHANES dataset where dietary consumption behavior was measured via an interview. This method is subject to recall bias as the participant may not accurately remember all foods consumed, including their yogurt/dairy consumption, on the “average” day. Recall bias could also have been impacted by significant memory impairment, but the NHANES screening procedures for DSST excluded participants who were unable to understand how to perform the test, though this would likely bias the results towards the null. Since eating healthy foods is socially desirable, subjects may also be tempted to over-report their consumption. Within the dietary questionnaire, the question about yogurt consumption was combined with other dairy products, including cottage cheese, ice cream, and milkshakes, so some of the participants who answered this question may not have eaten any yogurt. This, however, would tend to bias results toward the null. Since this was a US-based study, some of the dairy foods represented may not be generalizable to global populations where dairy is not a staple. However, worldwide, there are many non-dairy foods that undergo fermentation for consumption, including oats, wheat, soy, fruit juice, and vegetables, among many others, which can provide probiotics for those populations [78,79]. The global importance of fermented foods is currently being recognized, and initiatives are underway [80]. Furthermore, since data are cross-sectional and binary, we were unable to establish if there is a dose-dependent relationship or temporal association between yogurt/dairy consumption and cognitive function. Further research is needed to fully elucidate whether there is a causal role for probiotic foods such as yogurt in cognitive function. Future studies combining multiple NHANES cohorts, longitudinal NHANES cohorts, or randomized controlled trials are needed to better evaluate the link observed in this study.

This study uses a national sample that is properly weighed to be representative of the population of the US and has demonstrated that probiotic consumption is associated with improvements in cognitive function. Therefore, the information contained herein could be used to make better recommendations by organizations such as The US Departments of Agriculture and Health and Human Services that are often employed by dietitians at hospitals and nursing homes to improve outcomes in the elderly population [81]. Other countries around the world that have collected data on the dietary patterns in their populations may conduct analyses to explore the value of dairy in their inhabitants [82,83,84]. Given that the proportion of people over 70 is going to continue to increase in the upcoming decades, even modest improvements in cognitive function will have a large impact on quality of life and the expense of caring for these individuals in the long term [85,86,87]. The lifetime cost of care for a person with Alzheimer’s disease is in excess of USD 377,000 [88,89]. Recent estimates of care for people with MCI were nearly USD 34,000 per person per year and in excess of USD 50,000 in those with dementia, so low-cost interventions such as dietary modifications are of the utmost importance. Dietary modifications have been shown to improve memory in patients with mild cognitive impairment [90], and delaying the progression of mild cognitive impairment to dementia has been shown to save USD 10,000 per person in a recent analysis [91].

## 5. Conclusions

Our study highlights the importance of our food choices in many aspects of human health. More and more research supports the significance of communication between the enteric and central nervous systems via the production of bioactive metabolites influenced by gut probiotics and flora. Daily yogurt/dairy consumption may be an important dietary factor contributing to healthy cognitive function in older adults. More direct evidence is required to determine if this relationship is causal and if yogurt/dairy consumption can help maintain healthy cognition in the older population.

## Figures and Tables

**Figure 1 nutrients-16-03631-f001:**
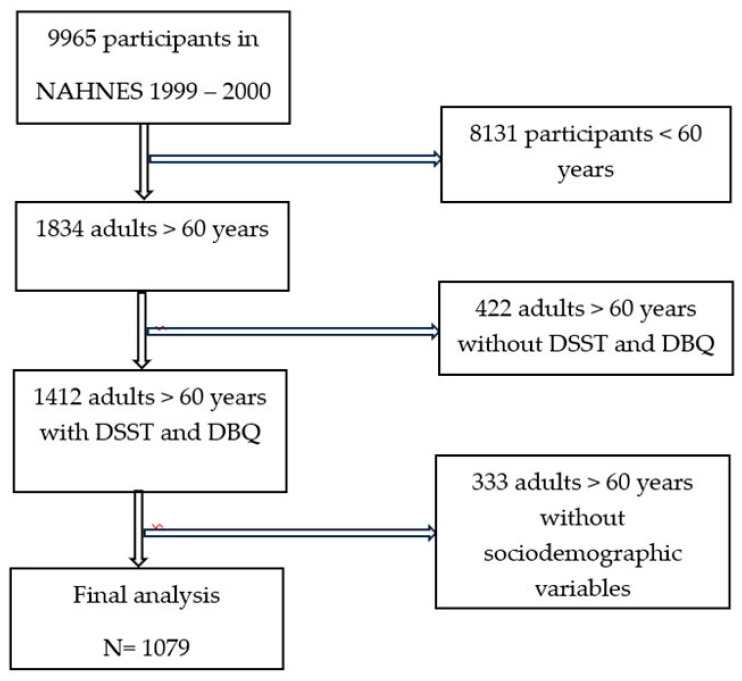
Flow chart of study participants. Abbreviations: NHANES: National Health and Nutrition Examination Survey; DSST: Digit–Symbol Substitution Test; DBQ: Diet Behavior Questionnaire.

**Figure 2 nutrients-16-03631-f002:**
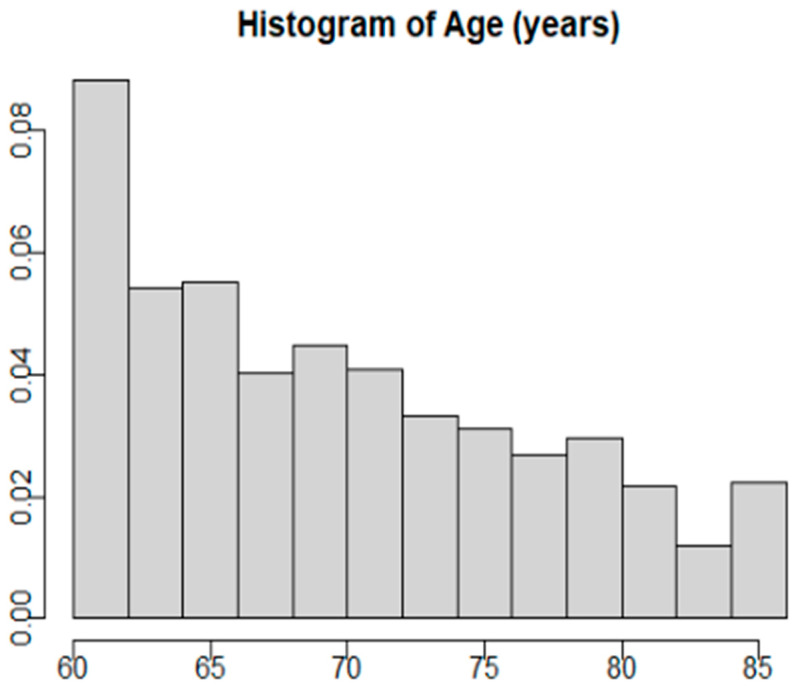
Histogram of weighted age in the final sample.

**Table 1 nutrients-16-03631-t001:** Weighted sociodemographic characteristics in the sample by yogurt consumption.

	No Yogurt/Dairy, N = 4,932,089	Daily Yogurt/Dairy, N = 25,158,064
Age (years)	69.13 (0.62)	70.28 (0.46)
Gender		
Male	2,247,781 (46%)	11,057,638 (44%)
Female	2,684,308 (54%)	14,100,426 (56%)
Race/Ethnicity		
Mexican American	152,947 (3.1%)	615,295 (2.4%)
Other Hispanic	363,091 (7.4%)	1,445,067 (5.7%)
Non-Hispanic White	3,777,027 (77%)	21,235,587 (84%)
Non-Hispanic Black	496,333 (10%)	1,238,224 (4.9%)
Other Race—Including Multi-Racial	142,690 (2.9%)	623,892 (2.5%)
Education Level		
Less than 9th grade	1,007,297 (20%)	3,194,640 (13%)
Grades 9–11	1,149,097 (23%)	4,225,745 (17%)
High School Grad/GED	1,267,298 (26%)	8,367,241 (33%)
Some College/Associates Degree	953,831 (19%)	5,248,348 (21%)
College Graduate or Above	554,565 (11%)	4,058,041 (16%)
Unknown	0.00	64,049.00
Poverty-to-Income Ratio	2.33 (0.17)	2.72 (0.15)
BMI (kg/m^2^)	28.49 (0.60)	28.09 (0.18)
Digit–Symbol (# correct)	41.45 (1.79)	46.19 (0.87)
*n* (%); Mean (SE)		

Abbreviations: BMI—body mass index; GED—General Education Diploma; Grad—graduate.

**Table 2 nutrients-16-03631-t002:** Weighted Regression of DSST Scores on Yogurt Consumption and Sociodemographic Variables.

	Unadjusted			Adjusted		
Characteristic	Beta	95% CI	*p*-Value	Beta	95% CI	*p*-Value
Yogurt Consumption						
No Yogurt/Dairy	—	—	—	—	—	—
Daily Yogurt/Dairy	4.74	0.98–8.50	0.017	2.43	−0.31–5.16	0.074
Age (years)	−0.80	−0.96–−0.64	<0.001	−0.61	−0.72–−0.51	<0.001
Education Level						
Less than 9th grade	—	—	—	—	—	—
Grades 9–11	13.40	8.82–17.99	<0.001	11.18	6.37–16.00	0.001
High School Grad/GED	23.17	19.10–27.24	<0.001	18.82	14.32–23.32	<0.001
Some College/Associate Degree	25.36	22.05–28.68	<0.001	19.44	15.64–23.23	<0.001
College Graduate or Above	30.06	25.39–34.72	<0.001	21.97	16.37–27.57	<0.001
Poverty-to-Income Ratio	5.25	4.52–5.98	<0.001	2.51	1.30–3.71	0.002

Abbreviations: CI—confidence interval; DSST—Digit–Symbol Substitution Test; GED—General Education Diploma; Grad—graduate.

**Table 3 nutrients-16-03631-t003:** Estimated cost savings per year in the US if transition from MCI to dementia could be reduced by 1%.

Diagnosis	Prevalence in US Persons Age 65+	Rate of Progression (% per Year)	Estimated Total Number of Persons in US with Diagnosis	Added Cost per Person per Year Compared to Cognitively Intact (USD)	Total Added Cost per Year Compared to Cognitively Intact (USD)	Savings per Person by Preventing MCI to Dementia Progression (USD)	Total Savings by Preventing 1% MCI from Progressing to Dementia (USD)	References
MCI	22.7 per 100		12,258,000	33,700	USD 413,094,600,000			[5]
MCI progressing to dementia		10	1,225,800					[24,26,27]
Dementia	4 per 100		2,160,000	51,000	USD 110,160,000,000	17,300	USD 212,063,400	[28]

Abbreviations: United States (US); mild cognitive impairment (MCI).

## Data Availability

The original contributions presented in the study are included in the article. Further inquiries can be directed to the corresponding author. The datasets supporting the conclusions of this article are publicly available from the NHANES (https://www.cdc.gov/nchs/nhanes/index.htm accessed on 4 August 2023).

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
