# Peer review of "Cognitive Function and the Consumption of Probiotic Foods: A National Health and Nutrition Examination Survey Study"

_nutrients, 2024, doi:10.3390/nu16213631_

Round 1
Reviewer 1 Report (Previous Reviewer 1)
Comments and Suggestions for Authors
Dear Ms.Sofia Zhou,
Thank you for the revised version of the manuscript titled "Cognitive function and the consumption of probiotic foods: an NHANES study." I have reviewed the revisions and I am pleased to accept them. I recommend the manuscript for publication.
Thank you,
Mónika Fekete, MD, PhD
Author Response
We very much appreciate the reviewer’s thorough scrutiny of our manuscript.
Below, we provide a point-by-point response to the reviewer’s comments.
Reviewer # 1 Comments and Responses
REVIEWER 1
Reviewer #1 Comment 1: Thank you for the revised version of the manuscript titled "Cognitive function and the consumption of probiotic foods: an NHANES study." I have reviewed the revisions and I am pleased to accept them. I recommend the manuscript for publication.
Response: We thank the reviewer for these kind words.

Reviewer 2 Report (New Reviewer)
Comments and Suggestions for Authors
Manuscript Comments (nutrients-3253224)
This work aims to determine whether the consumption of probiotics is related to increased measures of cognitive function in a representative national sample. The author analyzed the correlation between probiotic intake and mental improvement in the MCI population through extensive research data screening. The research results can help people understand the role of probiotics in cognitive function improvement. The following are some suggested modifications that should be double-checked before being considered for publication by Nutrients.
Detail comments:
1. L37-41: Suggest the author supplement relevant data or information on MCI worldwide.
2. L181: “kg/m2”, not “kg/m2” in Table 1.
3. L315-322: Whether the author put forward some reasonable suggestions for the diet of the global MCI population, not just for the American population?
4. References (L419-420 and L534-535) format is incorrect.
Comments on the Quality of English LanguageMinor editing of English language required.
Author Response
We thank the reviewer for thoroughly scrutinizing our manuscript. As requested, we have revised the manuscript and addressed the specific comments of the reviewer. The revised sections are delineated in red in a marked copy of the manuscript text.
Below, we provide a point-by-point response to the reviewer’s comments.
Reviewer # 2 Comments and Responses
REVIEWER 2
Reviewer #2 Comment 1: L37-41: Suggest the author supplement relevant data or information on MCI worldwide.
Response: Thank you, we have added information on global MCI prevalence with references, lines 41-43 and new references 7 and 8.
Reviewer #2 Comment 2: L181: “kg/m2”, not “kg/m2” in Table 1.
Response: We have made this correction.
Reviewer #2 Comment 3: L315-322: Whether the author put forward some reasonable suggestions for the diet of the global MCI population, not just for the American population?
Response: Thank you, we have added information on non-dairy fermented foods accessible to global populations, lines 311-317 and references 79-81.
Reviewer #2 Comment 4: References (L419-420 and L534-535) format is incorrect.
Response: We have fixed the formatting.
We thank the reviewers and believe that the manuscript is improved as a result of their input. We hope you will agree, and decide in favor of accepting our report at this time.

This manuscript is a resubmission of an earlier submission. The following is a list of the peer review reports and author responses from that submission.
Round 1
Reviewer 1 Report
Comments and Suggestions for Authors
The manuscript is well-written, the statistics are accurate, and this article contributes to a more comprehensive understanding of the effects of probiotics on cognitive health. However, further longitudinal studies are necessary to confirm the results, as the authors also acknowledge. The use of NHANES 1999-2000 data is appropriate (though recall bias is possible). The conclusions of the article are important as they highlight a potentially simple and cost-effective intervention for improving cognitive health in the elderly population, and the results are consistent with previous studies. The manuscript contains no plagiarism, the English is simple and clear, and the references are relevant.
It is recommended to include an abbreviation list under the tables (e.g., BMI, GED, SE, GRAD, etc.) and briefly describe the statistical methods and significance levels used, so the tables are understandable on their own. In the table, it would be useful to indicate (n, %) at the top or in the row, and there's no need to write out the % sign everywhere.
For original articles, it is advisable to use a structured abstract with sections such as Introduction, Objectives, Methods, Results, and Conclusions. The article is concise, straightforward, understandable, and short, making it easy for readers to process.
Author Response
We thank the reviewer for thoroughly scrutinizing our manuscript. As requested, we have revised the manuscript and addressed the specific comments of the reviewer. The revised sections are delineated in red in a marked copy of the manuscript text.
Below, we provide a point-by-point response to the reviewer’s comments.
Reviewer # 1 Comments and Responses
Reviewer #1 Comment 1: The manuscript is well-written, the statistics are accurate, and this article contributes to a more comprehensive understanding of the effects of probiotics on cognitive health. However, further longitudinal studies are necessary to confirm the results, as the authors also acknowledge. The use of NHANES 1999-2000 data is appropriate (though recall bias is possible). The conclusions of the article are important as they highlight a potentially simple and cost-effective intervention for improving cognitive health in the elderly population, and the results are consistent with previous studies. The manuscript contains no plagiarism, the English is simple and clear, and the references are relevant.
Response: We thank the reviewer for these kind words.
Reviewer #1 Comment 2: It is recommended to include an abbreviation list under the tables (e.g., BMI, GED, SE, GRAD, etc.) and briefly describe the statistical methods and significance levels used, so the tables are understandable on their own. In the table, it would be useful to indicate (n, %) at the top or in the row, and there's no need to write out the % sign everywhere.
Response: We have incorporated an abbreviation list and described statistical methods and added n, % while omitting % sign throughout.
Reviewer #1 Comment 3: For original articles, it is advisable to use a structured abstract with sections such as Introduction, Objectives, Methods, Results, and Conclusions. The article is concise, straightforward, understandable, and short, making it easy for readers to process.
Response: This is not within our control as the Author Instructions state clearly that: “The abstract should be a total of about 200 words maximum. The abstract should be a single paragraph and should follow the style of structured abstracts, but without headings.”
We thank the reviewer and believe that the manuscript is improved as a result of their input. We hope you will agree, and decide in favor of accepting our report at this time.

Reviewer 2 Report
Comments and Suggestions for Authors
This paper analyses in detail the correlation analysis between dairy products/yogurt and cognitive impairment in elderly people. Using multiple data analysis methods to analyze a large sample size, it was found that dietary changes (regular intake of dairy products/yogurt) were able to protect against cognitive decline in the elderly. However, further in-depth analyses and discussions about dairy-induced changes in gut flora are still needed.
1. Section 2.1. Probiotic consumption behavior is determined just by the question of whether Yogurt consumption is yes or no. There was no information about what bacterial species-based yogurt was consumed by each sample/group. It should be classified.
2. Section 2.2. Cognitive functions. The test factors should be explained.
3. The research mentioned that 1079 have information that can be analyzed, is the sample size sufficient for analysis?
4. It is recommended that a histogram of the normal distribution of participants' ages be added to better understand the interval distribution of participants' ages.
5. This study does not explain in detail the changes in intestinal flora caused by the ingestion of dairy products, so please add this to the discussion.
6. Additional relevant data analyses or in-depth discussions are recommended regarding cognitive alterations induced by gut flora.
7. The probiotic species suggested for yogurt or cheese need to be listed in detail in the manuscript for a clearer understanding of the role probiotics play.
8. The conclusion section is simply too short and could be further summarized in the results section.
Comments on the Quality of English Language
Minor editing of English language required.
Author Response
We thank the reviewer for thoroughly scrutinizing our manuscript. As requested, we have revised the manuscript and addressed the specific comments of the reviewer. The revised sections are delineated in red in a marked copy of the manuscript text.
Below, we provide a point-by-point response to the reviewer’s comments.
Reviewer # 2 Comments and Responses
Reviewer #2 Comment 1: Section 2.1. Probiotic consumption behavior is determined just by the question of whether Yogurt consumption is yes or no. There was no information about what bacterial species-based yogurt was consumed by each sample/group. It should be classified.
Response: This was not possible to report because the NHANES dietary recall questionnaire did not ask for specific yogurt type consumed by participants.
Reviewer #2 Comment 2: Section 2.2. Cognitive functions. The test factors should be explained.
Response: We have now included an explanation of the cognitive function test used and an additional reference (lines 88-91).
Reviewer #2 Comment 3: The research mentioned that 1079 have information that can be analyzed, is the sample size sufficient for analysis?
Response: Yes, based on a power calculation using our final model, 1079 participants is more than sufficient to achieve power >0.80 (Linear Regression (F test)
R-squared Deviation from 0 (zero) H0: r2 = 0 HA: r2 > 0 ------------------------------ Statistical power = 0.8 n = 24 ------------------------------ Numerator degrees of freedom = 4 Denominator degrees of freedom = 18.696 Non-centrality parameter = 15.342 Type I error rate = 0.05 Type II error rate = 0.2
Reviewer #2 Comment 4: It is recommended that a histogram of the normal distribution of participants' ages be added to better understand the interval distribution of participants' ages.
Response: A histogram has been added (Figure 1).
Reviewer #2 Comment 5: This study does not explain in detail the changes in intestinal flora caused by the ingestion of dairy products, so please add this to the discussion.
Response: The discussion has been expanded as requested.
Reviewer #2 Comment 6: Additional relevant data analyses or in-depth discussions are recommended regarding cognitive alterations induced by gut flora.
Response: This has been added to the discussion.
Reviewer #2 Comment 7: The probiotic species suggested for yogurt or cheese need to be listed in detail in the manuscript for a clearer understanding of the role probiotics play.
Response: The probiotic species have been incorporated with context in the discussion.
Reviewer #2 Comment 8: The conclusion section is simply too short and could be further summarized in the results section.
Response: The conclusion is presented in greater depth.
We thank the reviewer and believe that the manuscript is improved as a result of their input. We hope you will agree, and decide in favor of accepting our report at this time.
